# The Influence of Sociodemographic Characteristics and the Experience of Recreational Divers on the Preference for Diving Sites

Ke Zhang [1], Anson T. H. Ma [1], Theresa W. L. Lam [1], Wei Fang [1] and Lewis T. O. Cheung [1,2,*]

1   Department of Social Sciences, The Education University of Hong Kong, Hong Kong, China
2   Resource Centre for Interdisciplinary Studies and Experiential Learning, Faculty of Liberal Arts and Social Sciences, The Education University of Hong Kong, Hong Kong, China
*   Correspondence: ltocheung@eduhk.hk

**Abstract:** Understanding the preferences of scuba divers is crucial for authorities to establish appropriate management plans and for service providers to offer a wide range of recreation opportunities to promote long-term industrial sustainability. This study categorized diving preferences into two subgroups (physical and biological) to explore the association between diving preferences and divers' sociodemographic characteristics, as well as diving experiences through multiple regression analysis. Questionnaire surveys were administered, with 398 completed responses successfully collected from diving hotspots in Hong Kong. Results revealed that divers' sociodemographic status, particularly monthly salary, could affect their physical and biological preferences. Meanwhile, divers' diving experience was found to be a significant predictor in figuring out their biological preferences. However, no relationship between diving experience and physical preference was identified in this study. These results could inform management authorities in implementing ecological and environmental protection because biological conditions are considered the most attractive motivation for divers to dive in Hong Kong. Similarly, these results may help managers better understand divers' recreation needs based on different sociodemographic characteristics and diving experiences to create diverse recreation opportunities while enhancing their satisfaction by avoiding the negative impacts associated with identified preference attributes.

**Keywords:** scuba diving; biological preferences; physical preferences; diving experiences; Hong Kong

## 1. Introduction

With 7.4 million people living in a densely populated city [1], Hong Kong has experienced a remarkable social transformation in economic, cultural, and tourism prosperity over the past decades. Despite the intricate land problems that remain unsolved, Hong Kong has retained a high percentage of protected areas, with 1098 square kilometres accounting for 38% of the total land area, making Hong Kong the most protected area among all administrative regions in the Asia-Pacific region [2]. Protected areas in Hong Kong are mainly divided into three categories, namely 24 country parks, 22 special areas, and 7 marine parks covering 44,300 hectares of protected areas and 8500 hectares of marine protected areas [3]. In addition to the well-established protected areas, the government has strict planning and control over approximately 7700 hectares of additional land by taking advantage of its existing resources to enhance environmental conservation, recreation opportunities, and scientific development to prepare society to achieve long-term sustainable prosperity [4].

However, marine-protected areas are not entirely off-limits to visitors in Hong Kong. This condition has steadily promoted a wide range of water-based recreational activities, strengthening human well-being and resulting in positive recreational experiences for

recreationists [5]. In particular, scuba diving is recognized as one of the most popular water-based activities for people visiting marine protected areas. Hoi Ha Wan, Yin Tsz Ngam, Ung Kong Chau (Bluff Island), Ninepin Islands, and Sharp Island are consistently popular destinations for snorkelling and scuba diving in Hong Kong [6]. Previous research has suggested that divers' selection of dive sites varies depending on their sociodemographic characteristics and diving experience. For instance, Rouphael and Inglis [7] discovered that male divers prefer to dive deeper than female divers, while Emang et al. [8] revealed that experienced divers appreciate marine biodiversity more than divers with less diving experience. However, little is known about who dives and why divers select these protected areas for recreational diving, especially with regard to divers' diving preferences in Hong Kong. Although scholars have conducted studies on resource management [9,10], destination marketing [11,12], and expectations [13–15] to understand divers' preferences, very little literature has explored nature-based scuba diving preferences in Hong Kong and in the Chinese community. Moreover, previous literature concerning divers' diving preferences has relied heavily on psychological criteria such as perception [16–18], motivation [19–21], and conservation attitude [22,23]. In contrast, divers' sociodemographic status and diving experiences as critical internal factors have been overlooked or rarely explored [16]. Thus, understanding recreational divers' diving experiences and preferences are crucial for helping tourism businesses develop new market strategies to make their businesses more competitive. Meanwhile, it may also provide management authorities with practical scientific advice to ensure that environmental resources and tourist experiences are not compromised.

This study aims to investigate the relationship between divers' sociodemographic characteristics, experience, and specific destination preferences while researching how diving experience influences divers' preferences. Specifically, three research questions have been developed to address the research objectives. (1) What is the relationship between recreational divers' sociodemographic characteristics and diving experiences? (2) What is the relationship between recreational divers' sociodemographic characteristics and diving site preferences? (3) What is the relationship between recreational divers' diving experiences and diving site preferences? With the research objectives in mind, the proposed study is anticipated to provide evidence on the correlation between the sociodemographic characteristics of recreational divers, diving experience, and specific diving preferences (physical and biological). More importantly, the results of this study will contribute to implementing tourism management decisions while supporting scuba diving to move towards a more sustainable nature-based tourism industry.

## 2. Literature Review

### 2.1. Marine Protected Areas and Scuba Diving

Marine protected areas have proven to be one of the most effective management strategies in response to resource depletion and conservation pressure, in addition to their well-known socioeconomic, environmental education, and scientific research benefits [24–27]. In particular, the abundant wildlife resources and stunning natural landscapes preserved in MPAs have expanded recreational opportunities while promoting cultural benefits, such as creating alternative employment opportunities and improving aesthetic values [24,27]. For these reasons, scholars have widely studied recreational activities as an unavoidable component of nature-based tourism, including snorkelling [28,29], scuba diving [5,16,30–32], and fishing [33,34] in protected marine areas. Among them, scuba diving has undoubtedly become one of the most popular water-based recreational activities due to its commercial success, capacity to relieve stress and encourage mental relaxation, and physical health advantages [35–37].

As a result, dive-related films and documentaries, commercial manufacturing, and dive training programmes are constantly increasing in the US [38]. Likewise, approximately 6600 PADI dive shops and resorts have been established worldwide due to the tremendous expansion in the number of certified divers and the rapidly growing demand

for recreational activities [39], making diving tourism a vital part of global economic development [32]. Since its inception, scuba diving has benefited tourists and host communities, allowing individual divers to relax physically and mentally [35,36] while host communities enjoy significant economic benefits from diving business operations [37,40]. However, some negative aspects have also been reported with the increased popularity of recreational scuba diving in the MPAs [41,42]. For instance, studies have demonstrated that inappropriate underwater diving behaviour for recreational divers could severely damage aquatic species and marine ecosystems [30,43]. In addition to inappropriate diving behaviour, a study noted that the increased usage of marine protected areas for scuba diving might further diminish the recreational value of dive sites while may progressively reduce tourists' recreational satisfaction [44]. As such, implementing an appropriate management plan that balances visitors' recreational experience and environmental preservation is crucial for the long-term growth of sustainable scuba diving tourism.

### 2.2. The Influence of Sociodemographic Variables on Diving Experience

Diving experiences, as a critical element of scuba diving, have been extensively discussed, but a complete understanding of diving experiences is as crucial as environmental precautions in maintaining the scuba diving tourism industry [5]. Studies have presented the definitions and measurements of diving experience from various perspectives. For instance, Lucrezi et al. [45] indicated that the number of dives determines diving experience, followed by familiarity with diving sites due to divers' past participation. However, Ranapurwala et al. [46] reported that diving experience is challenging to measure and that regularly tracking the annual number of dives a diver makes is more accurate than counting the years that the diver has been diving. Similarly, divers' diving experience has been measured using indicators such as the number of dives undertaken in a lifetime, certification level, or self-rating, which has been widely applied throughout the scuba diving context worldwide [8,16,31,45].

Therefore, understanding divers' experiences and their impacts may benefit the management of dive destinations across the globe. This requires assessing whether diving experience has a consistent impact across sites and various groups of divers [45]. Under those circumstances, diving experiences related to tourism issues continue to be explored by scholars in response to the ongoing expansion and management needs of scuba diving activities. A survey of 302 divers in Malaysia reported that diving experience is highly dependent on the sociodemographic background of recreational divers [31]. The author revealed that males obtain more diving experience than females, and older and better-educated divers perceive more diving experience than younger divers and those who only receive a diploma. In the same study, the author further discovered that different ethnicities of scuba divers significantly differ in self-rating their level of diving experience. Particularly, divers of other ethnicity and the ethnicity of Malaysian Chinese consider themselves more experienced than those of Malay and Indian ethnicity when diving in Malaysia. Likewise, an evaluation of the diving experience of French recreational divers found that gender greatly influenced their diving experience [47]. Men had a higher level of diving experience compared to women. In addition to the general recreation experiences, Ranapurwala, Bird, Vaithiyanathan, and Denoble [46] found that divers' injury status was highly associated with their sociodemographic background. Men reported fewer diving injuries than women, while divers with instructor-level certifications received fewer injuries than those with basic diving certifications. However, only a few studies have compared the differences in divers' diving experience based on divers' sociodemographic characteristics, and even fewer have been applied to the Chinese context. Given the current literature development, the first hypothesis is proposed.

**Hypothesis 1 (H1).** *There is a relationship between divers' sociodemographic characteristics and diving experiences.*

*2.3. The Influence of Demographic Variables on Diving Preferences*

Divers' preferences are a complicated study topic that has been investigated in various locations through multiple indicators to understand and explain divers' motivations for the selection of diving sites. Previous studies have demonstrated that most recreational divers choose dive locations with a high concentration of corals, unusual fish, and turtles [18]. Comparatively, muck divers prefer dive sites with a high concentration of blue-ringed octopus, flamboyant cuttlefish, and frogfish [48]. In addition to biological preferences, weather conditions, underwater visibility, popularity, surface condition, entry points, water quality, litter pollution, and divers' density have been recognized as critical environmental attributes for divers' site preferences [8,17,18,49–51]. However, the fact remains that scuba divers are not a homogeneous group, and divers' preferences might differ across individuals [16,17]. For instance, Meisel-Lusby and Cottrell [13] found significant differences in predicting the diving preferences of Scouts versus regular US divers in youth adventure programmes. Scout divers dive for the excitement of the adventure, while regular US divers dive for the purpose of using diving equipment. In addition, Şensurat-Genç, Shashar, Özsüer, and Özgül [23] indicated that gender influenced dive preference for particular physical environmental attributes of the destination. Men were more likely to enjoy wrecks than women, while women preferred sites with more spaced locations.

Likewise, Uyarra, Watkinson, and Cote [18] found that gender influenced dive preferences for perceiving differences between fish and coral conditions. Male divers perceive diving preferences more based on coral qualities with regard to percentage coverage, structural complexity, and coral species richness. In contrast, female divers generate diving preferences more based on the abundance of a shoal of fish. Also, Rouphael and Inglis [7] revealed that male divers tend to be more adventurous, preferring to dive deeper and explore caves to increase their likelihood of encountering large fish than female divers. However, little information is available regarding how divers' diving preferences differ according to demographic status among the marine protected areas in Hong Kong. This topic needs to be promoted to improve the management of the diving industry and determine its potential effects on the marine environment. Based on these studies, the second hypothesis is proposed.

**Hypothesis 2 (H2).** *There is an association between divers' sociodemographic characteristics and diving preferences.*

*2.4. Diving Experiences and Diving Preferences*

Studies on divers' diving experiences and preferences have been conducted worldwide, including in Malaysia [8,48], Spain [15], Brazil [5,16], Barbados [17], Israel [52], Portugal [53], the United States [13], and South Africa [20]. Previous studies have indicated that understanding diving experience is vital for managing marine protected areas, and divers' diving preferences are significantly impacted by the levels of diving experience they have acquired [5,16,20,54]. For instance, Giglio, Luiz, and Schiavetti [16] found that male and female divers' perceptions of the megafauna and cryptic animals changed significantly with increasing experience through interviewing 190 recreational divers at the Abrolhos National Marine Park in eastern Brazil. Particularly, experienced divers preferred encounters with cryptic fishes and small invertebrates, while novice divers preferred to enjoy giant sea creatures such as sharks and rays. Emang, Lundhede, and Thorsen [8] conducted a choice experiment analysis on 507 recreational scuba divers at Sipadan, Borneo in Malaysia. The authors found that experienced divers had a much stronger appreciation for marine biodiversity, while divers with less experience perceived a lower probability of seeing pelagic species.

In addition to the marine biological environment, Kirkbride-Smith, Wheeler, and Johnson [17] claimed that novice divers found artificial reefs highly enjoyable. In contrast, experienced divers were more inclined towards natural reefs as they became more

experienced. Moreover, Fitzsimmons [51] found that novice diver satisfaction features were substantially more driven by equipment and individual diving abilities, while the ecological diversity of social aspects more determined experienced divers. Conversely, Tynyakov, Rousseau, Chen, Figus, Belhassen, and Shashar [52] revealed that recreational divers like to switch up with dive sites to achieve the desire to have a diverse and expanded diving experience. The author further emphasizes that divers showed a great willingness to dive on artificial reefs regardless of their diving experience. Interestingly, although the more experienced divers were motivated to find new species, the less experienced divers persisted in overcoming their fears and learning new diving skills to prepare to explore new dive spots [20]. Given the information in the literature, the third hypothesis is proposed.

**Hypothesis 3 (H3).** *There is an association between divers' diving experience and diving preferences.*

### 3. Methodology

#### 3.1. Study Areas

Hong Kong is located on the southern coast of China with a small land area of 1106 square kilometres alongside 1649 square kilometres of water area, making Hong Kong a total area of 2755 square kilometres [55]. In particular, Hong Kong has a subtropical climate with a maximum annual average temperature of about 27.5 degrees and an annual minimum temperature of 22.6 degrees Celsius [56]. Its geographical and climatic advantages have contributed to a rise in the tourist industry, particularly in areas that focus on nature-based tourism, such as swimming, snorkelling, and scuba diving. These water activities generally take place in the eastern part of the Marine Reserve, where Hoi Ha Wan Marine Park, Tung Ping Chau Marine Park, Sharp Island, and the Ung Kong Group are the most famous traditional dive sites in Hong Kong. Specifically, these sites have relatively widespread rocky hard bottoms with more oceanic water conditions that allow the development of expansive coral ecosystems and marine species [57], which have played an important role in attracting most scuba divers [58]. Therefore, to ensure representative sampling is achieved, Hoi Ha Wan Marine Park, Tung Ping Chau Marine Park, Sharp Island, and the Ung Kong Group were the study sites to invite eligible divers to participate in facilitating the data collection.

#### 3.2. Questionnaire Design

A questionnaire was developed and used as the primary research instrument for this study. The questionnaire was divided into three sections. The first section collected demographic variables, including gender, age, educational background, and monthly income. The second section explored divers' diving experience as characterized by "years of first diving qualification," "number of dives with acquired qualification," "the highest level of diving qualification," and "number of diving places outside of HK," which were adapted from previous studies [8,16,30,31,45,59] with slight modifications to comply with the diving context in Hong Kong. The last section studied divers' diving preferences in terms of biological and physical site environmental conditions, including the "level of naturalness and pristineness of dive sites," "biodiversity and ecological value," "uniqueness of dive sites," "water quality and clarity," "Site area," and "coral density and coverage," which were adapted from previous works in the literature following our current research objectives [15,16,18,20,50,51,60]. Considering the proposed methodology, the demographic and dive experiences variables were examined using closed questions with a checklist. At the same time, a five-point Likert scale was used to evaluate diving site preferences from "very low" (1) to "very high" (5).

### 3.3. Sampling Method

We collected questionnaires around dive sites in Hoi Ha Wan Marine Park, Tung Ping Chau Marine Park, Sharp Island, and the Ung Kong Group (Figure 1) in the summer of 2021 and 2022. This study employed a convenience sampling method because of its significant advantages of being less expensive and time-consuming for researchers [61], especially given the current pandemic-caused social distancing restriction orders in Hong Kong. However, an attempt was made to target potential respondents with more efficient computerized questionnaire distribution strategies to improve the sample's representativeness. Specifically, the research questionnaires were converted into QR codes and distributed to local dive shops, instructors, and commercial vendors near the dive sites to facilitate the search for potential respondents. Individuals who responded to this questionnaire were over 18 years old and had previous scuba diving experience. The reason for adopting a web-based data collection strategy is that online recruitment offers significant advantages, including low cost, time savings, and more accessible data storage and visualization than traditional questionnaire distribution [62]. In addition, a web-based survey method allows for cost-effective data collection and rapid access to dive tourists throughout Hong Kong to obtain a broad sampling distribution [59], especially given the shortcomings of data collection during the uncertain pandemic outbreak. A total of 550 questionnaires were successfully distributed, and approximately 420 questionnaires were returned. The actual number of valid questionnaires was counted as three hundred and ninety-eight (N = 398), with a response rate of 76% after censoring and eliminating incomplete or invalid questionnaires.

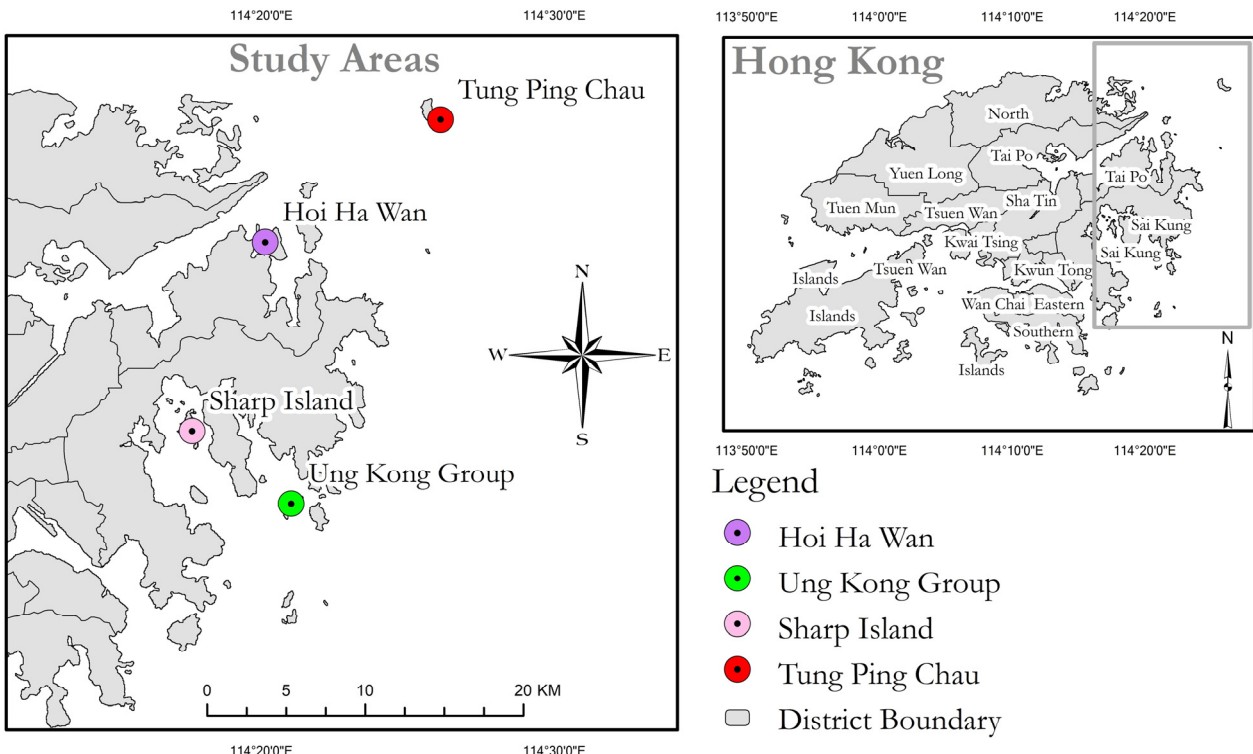

**Figure 1.** Study areas: Hoi Ha Wan, Tung Ping Chau, Sharp Island, and Ung Kong Group MPA.

### 3.4. Data Analysis

All data obtained from the questionnaire were evaluated with normal distribution analysis using SPSS 27.0 to establish the direction of the subsequent data analysis according to the research questions. Once it was confirmed that the data did not fail the normal distribution test, descriptive statistics and multiple linear regression analysis were conducted as parametric statistics to examine the relationship between the independent and dependent variables in light of the study objectives.

## 4. Results

### 4.1. Data Normality

With the designated methodology in mind, the normality test was performed before executing the parametric tests for the multiple regression analysis [63]. In fact, numerous statistical procedures have used normality assumptions for quantitative research methods, such as correlation, regression, t-test, and ANOVA. Typically, the mean value can reasonably represent the data set and assess the degree of significance (*p*-value) when the data are normally distributed [63]. Conversely, the resultant mean may not be reasonable to represent the complete information data set, which may result in an inaccurate interpretation with different research outputs. As such, researchers have identified a variety of approaches to validate the data distribution, which can be roughly classified into two types: "numerical tests" and "graphical interpretations" [63]. Specifically, the Shapiro–Wilk and Kolmogorov–Smirnov tests are used for small or medium sample sizes, while the eyeball test is helpful for medium or large sample sizes [64]. By contrast, graphical interpretation is an excellent way to assess normality when numerical tests may be overly sensitive or insensitive [63].

Given the limitations of several methods in conjunction with our respondent sample size (398), the skewness and kurtosis distribution methods were chosen to determine normality by applying the z-test calculation method with regard to their adaptability and flexibility for studying different sample sizes [64]. When the absolute z value belongs to the range of $\pm 3.29$, the data set is normally distributed [64]. Correspondingly, the skewness and kurtosis of all 14 indicators for the observation of the current study were under $\pm 3$, which indicated that our data were normally distributed (Table 1).

**Table 1.** Data of normality analysis.

| Measurement Variables | Skewness | Kurtosis |
|---|---|---|
| Gender | 0.065 | −2.006 |
| Age | 0.970 | 0.185 |
| Level of education | −0.037 | −0.634 |
| Monthly base salary HKD | 0.872 | −0.345 |
| Years since first diving qualification | −0.796 | −0.495 |
| Number of dives achieved | 0.417 | −1.018 |
| Highest level of diving qualification | 0.564 | −0.721 |
| Number of diving places outside of HK | 0.633 | −0.687 |
| Naturalness/pristineness | −0.326 | −0.322 |
| Biodiversity/ecological | −0.685 | 0.058 |
| Uniqueness | −0.155 | −0.330 |
| Water quality and clarity | −0.320 | −0.943 |
| Coral density/coverage | −0.319 | −0.377 |
| Site area | 0.343 | 0.310 |

### 4.2. Respondents' Demographic Characteristics and Response Rate

Prior to conducting the follow-up statistical data analysis, 398 completed questionnaires were obtained. Detailed information concerning the sociodemographic characteristics of the respondents is presented in Table 2. Regarding gender distribution, 193 of the respondents were male, and 205 were female, representing 48.5% and 51.5% of the total respondents, respectively. With regard to age, almost half of the respondents were in the 18–29 age group (45.7%), followed by 123 divers in the 30–39 age group (31%); only 31 divers (7.8%) were 50 or above. Based on the level of education, undergraduate students accounted for the majority of respondents with 229 divers (57.5%), followed by graduates and secondary school students with 92 divers (23.1%) and 77 divers (19.3%), respectively. The respondents' average monthly wage was computed and examined: 104 divers (26.1%) earned a salary between HKD 20,000 and HKD 29,999 per month, followed by 92 divers (23.1%) who earned an average wage of HKD 10,000–19,999 per month. Notably, 24 (6%)

respondents made less than HKD 9999 per month, while 52 respondents were either retired or did not divulge their wage status.

**Table 2.** Respondents' sociodemographic characteristics.

|  | **Frequency** | **%** |  | **Frequency** | **%** |
|---|---|---|---|---|---|
| Gender |  |  | Salary/Month ($HK) |  |  |
| Male | 193 | 48.5 | 9999 or below | 24 | 6.0 |
| Female | 205 | 51.5 | 10,000–19,999 | 92 | 23.1 |
|  |  |  | 20,000–29,999 | 104 | 26.1 |
| Age |  |  | 30,000–39,999 | 54 | 13.6 |
| 18–29 | 182 | 45.7 | 40,000–49,999 | 32 | 8.0 |
| 30–39 | 123 | 30.9 | 50,000–59,999 | 17 | 4.3 |
| 40–49 | 62 | 15.6 | 60,000 or above | 23 | 5.8 |
| 50–59 | 27 | 6.8 | Retire | 26 | 6.5 |
| 60 or above | 4 | 1.0 | Do not answer | 26 | 6.5 |
|  |  |  | Total | 398 | 100 |
| Education |  |  |  |  |  |
| Primary | 0 | 0 |  |  |  |
| Secondary | 77 | 19.3 |  |  |  |
| Undergraduate | 229 | 57.5 |  |  |  |
| Postgraduate | 92 | 23.1 |  |  |  |

### 4.3. Diving Experience

This section summarizes the information regarding the respondents' diving experience (Figure 2). The "timeline for qualification as a qualified diver" demonstrated that 82.1% of divers had been qualified for at least one year, followed by 36.6% who had been certified for more than five years or longer before participating in this research. In terms of "numbers of dives with acquired qualification", more than two-thirds (76.1%) of divers had completed at least 20 dives, while almost half (45.5%) of divers had completed more than 100 dives. Following closely behind the numbers of dives was "the degree of diving qualification", with 39.2% of divers who had advanced open water certification or above; over half (46.7%) of the divers were qualified as rescue divers, dive masters, or dive instructors. Finally, divers' overseas diving experience in conjunction with their dive count revealed that more than 81.9% of divers had visited dive sites outside of Hong Kong, and nearly half (40.4%) of divers had obtained five or more overseas diving trip experiences. In comparison, only 18.1% of divers had no overseas diving experience.

### 4.4. Site Preference of Recreational Divers

Six items were divided into two preference variables, including the physical environment and biological preferences, referring to the previous literature in line with our research objectives [15,16,18,20,49,60]. The first variables of physical environment preference included "naturalness", "uniqueness", "water quality and clarity", and "site area". The second variables of biological environment preference included "biodiversity" and "coral density". Since we separated the diving preference measurements into two subgroups, a reliability test was performed to ensure that the measuring variables were consistently reliable. The results from the variable reliability testing are shown in Table 3. All variables were deemed reliable since they had Cronbach's alpha values greater than 0.6, indicating that the proposed variables were eligible to be selected for further analysis [65].

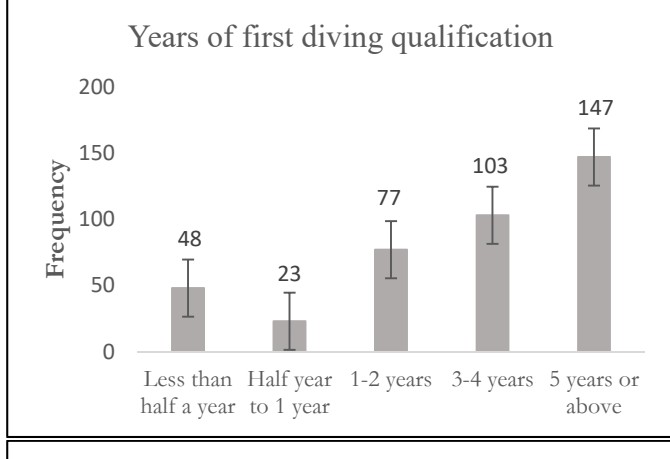
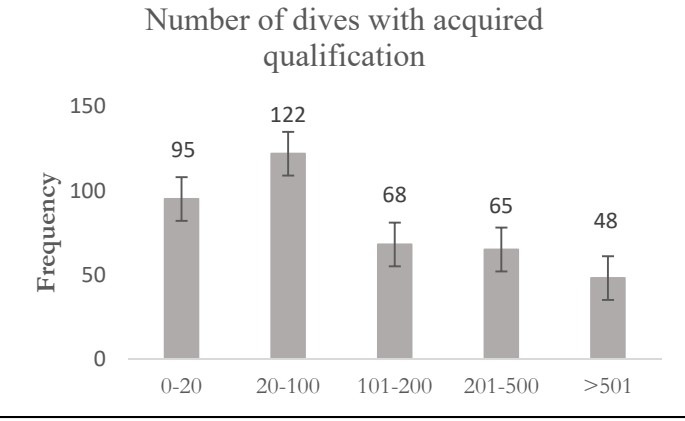
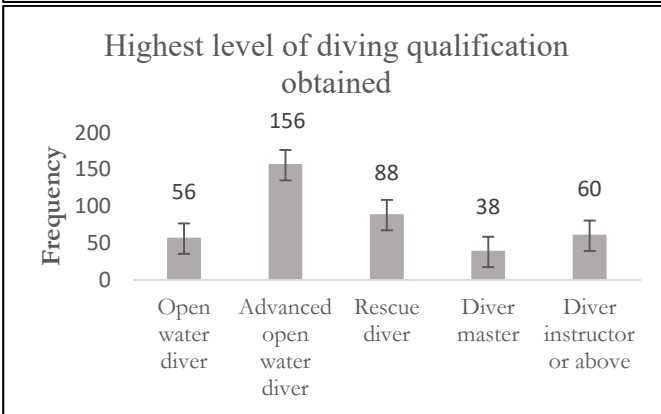
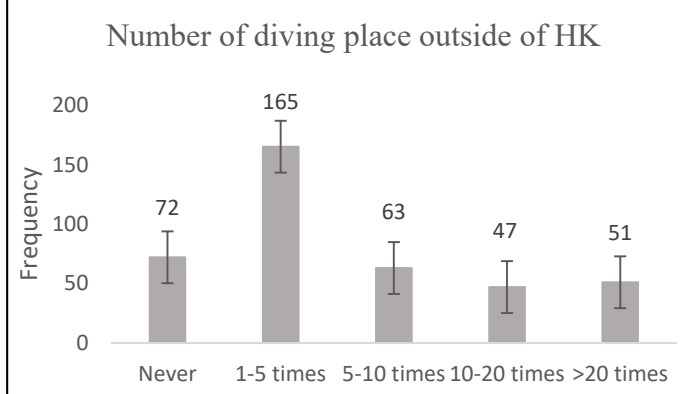

**Figure 2.** Diving experience of the respondents.

**Table 3.** Indicators of site preferences.

| Physical Preferences (α = 0.78) | Very Low | Low | Average | High | Very High | Mean |
|---|---|---|---|---|---|---|
| Naturalness/pristineness | 0.5 | 2.8 | 28.4 | 41.7 | 26.6 | 3.91 |
| Uniqueness | 0.8 | 5.3 | 36.4 | 38.2 | 19.3 | 3.70 |
| Water quality and clarity | 5.0 | 19.3 | 22.1 | 29.1 | 24.4 | 3.48 |
| Site area | Very small | Small | Average | Big | Very big | Mean |
|  | 0.3 | 8.0 | 57.3 | 29.1 | 5.3 | 3.31 |
| Biological preferences (α = 0.78) | Very low | Low | Average | High | Very high | Mean |
| Biodiversity/ecological | 1.0 | 3.8 | 22.4 | 37.2 | 35.7 | 4.03 |
| Coral density/coverage | 1.3 | 10.8 | 30.7 | 39.7 | 17.6 | 3.62 |

As reflected in Table 3, the most significant factors considered by divers concerning diving preferences were "biodiversity and ecological value", which received a mean score of 4.03 out of 5, followed closely by "degree of naturalness/pristineness" and "uniqueness", with mean scores of 3.91 and 3.70, respectively. Compared to the other variables in the analysis, the site area had the lowest mean score, with a mean score of 3.31. Therefore, it is reasonable to conclude that the site area is of little concern to scuba divers, whereas naturalness, originality, and ecological value are greater concerns for them. This result confirms previous work in which visibility, species diversity, and coral cover were essential factors influencing divers' decisions about diving site selection [8,13,15,16,50,51,60].

Preferences were rated on a five-point Likert scale from 1 for "very low" to 5 for "very high". The item for the site area was rated 1 for "very small" to 5 for "very big".

*4.5. Divers' Sociodemographic Characteristics and Physical and Biological Preferences*

The divers' demographic characteristics were considered independent variables, while diving experiences and preferences (physical and biological) were recognized as dependent

variables through three separate multiple linear regression analyses (Table 4). The first model identified the relationship between divers' sociodemographic status and diving experiences. Divers' sociodemographic characteristics could significantly predict their diving experience ($F$ (4,393) = 30.69, $p < 0.001$); the four predictors together could explain 24% of the total variance of diving experiences. Age ($b = 0.54$, $t = 9.78$, $p < 0.001$, $\beta = 0.47$) and education ($b = 0.30$, $t = 3.78$, $p < 0.001$, $\beta = 0.17$) were positively associated with divers' diving experience, whereas gender ($b = -0.00$, $t = -0.04$, $p = 0.97$, $\beta = -0.00$) and salary ($b = 0.02$, $t = 0.88$, $p = 0.38$, $\beta = 0.04$) did not significantly predict diving experiences. This indicates that older and better-educated Chinese divers may obtain more diving experiences than those younger and less-educated divers. The second model examined the relationship between divers' sociodemographic attributes and their physical preferences. Divers' demographic characteristics could significantly predict their physical preferences ($F$ (4,393) = 2.54, $p < 0.05$); the four predictors from the sociodemographic attributes together could explain 3% of the total variance of physical preferences. Monthly salary was a significant predictor of physical preferences ($b = -0.04$, $t = -2.41$, $p < 0.05$, $\beta = -0.13$), whereas gender ($b = 0.11$, $t = 1.55$, $p = 0.12$, $\beta = 0.08$), age ($b = 0.02$, $t = 0.51$, $p = 0.61$, $\beta = 0.03$), and education ($b = -0.04$, $t = -0.71$, $p = 0.48$, $\beta = -0.04$) did not significantly predict physical preferences, indicating that an increase in the monthly salary of recreational divers apparently leads to fewer physical preferences. The third model studied the relationship between divers' sociodemographic characteristics and their biological preferences. Divers' sociodemographic characteristics could also significantly predict divers' biological preferences ($F$ (4,393) = 2.60, $p < 0.05$); the four predictors from the sociodemographic attributes together could explain 3% of the total variance of biological preferences. Monthly salary was a significant predictor of biological preferences ($b = -0.05$, $t = -2.54$, $p < 0.05$, $\beta = -0.13$), whereas gender ($b = 0.15$, $t = 1.75$, $p = 0.08$, $\beta = 0.09$), age ($b = 0.00$, $t = 0.03$, $p = 0.97$, $\beta = 0.00$), and education ($b = 0.02$, $t = -0.30$, $p = 0.76$, $\beta = 0.02$) did not significantly predict biological preferences, indicating that an increase in the monthly salary of recreational divers apparently leads to fewer biological preferences concerning site selection.

**Table 4.** Coefficients of sociodemographic characteristics on divers' diving experience and physical and biological preferences.

| | Diving Experiences | | | | | Physical Preferences | | | | | Biological Preferences | | | | |
|---|---|---|---|---|---|---|---|---|---|---|---|---|---|---|---|
| | Unstandardized $b$ | Standardized Std. E | $\beta$ | $t$ | Sig. | Unstandardized $b$ | Standardized Std. E | $\beta$ | $t$ | Sig. | Unstandardized $b$ | Standardized Std. E | $\beta$ | $t$ | Sig. |
| Gender | −0.00 | 0.10 | −0.00 | −0.04 | 0.97 | 0.11 | 0.07 | 0.08 | 1.55 | 0.12 | 0.15 | 0.09 | 0.09 | 1.75 | 0.08 |
| Age | 0.54 | 0.06 | 0.47 | 9.78 | 0.00 *** | 0.02 | 0.04 | 0.03 | 0.51 | 0.61 | 0.00 | 0.05 | 0.00 | 0.03 | 0.97 |
| Education | 0.30 | 0.08 | 0.17 | 3.78 | 0.00 *** | −0.04 | 0.06 | −0.04 | −0.71 | 0.48 | 0.02 | 0.07 | 0.02 | 0.30 | 0.76 |
| Salary | 0.02 | 0.02 | 0.04 | 0.88 | 0.38 | −0.04 | 0.02 | −0.13 | −2.41 | 0.02 * | −0.05 | 0.02 | −0.13 | −2.54 | 0.01 * |
| R | | 0.49 | | | | | 0.16 | | | | | 0.16 | | | |
| R² | | 0.24 | | | | | 0.03 | | | | | 0.03 | | | |
| ΔR² | | 0.23 | | | | | 0.02 | | | | | 0.02 | | | |
| Std. Error | | 1.00 | | | | | 0.71 | | | | | 0.83 | | | |
| $df$ | | 4 | | | | | 4 | | | | | 4 | | | |
| $N$ | | 393 | | | | | 393 | | | | | 393 | | | |
| F-statistic | | 30.69 | | | | | 2.54 | | | | | 2.60 | | | |
| (F-statistic) | | 0.000 *** | | | | | 0.039 * | | | | | 0.036 * | | | |

Note: $p < 0.05$ *, $p < 0.001$ ***.

### 4.6. Relationship between Divers' Diving Experiences and Physical and Biological Preferences

Hierarchical multiple regression was used to assess the association between diving experience and two types of diving preferences (physical and biological). First, sociodemographic variables were set as control variables to ensure that they only explained some relationships between predictors and divers' site preferences. This was a concern because previous regression models had explained the relationship between divers' sociodemographic characteristics and their site preferences (Table 4). Second, the diving experience

was integrated into the hierarchical regression model to evaluate its predictive power in predicting divers' site preferences beyond demographic characteristics (Model 5). This measurement was based on statistical evidence of a positive relationship between divers' diving experience and diving preferences concerning site selection from several previous studies [5,8,16,17,29,48,51].

In particular, the coefficients, the analysis of variances, the F-statistics, and the $p$-values were investigated under the hierarchical regression model. In terms of divers' physical preferences, Table 5 indicated that the first model (diving experiences) could not accurately predict divers' physical preferences after controlling for the effects of sociodemographic variables because the measurement of diving experiences ($b = 0.06$, $p = 0.091$, $\beta = 0.10$) was not statistically significant at the level of 0.05. This indicates that diving experience does not affect divers' physical preferences in regard to selecting sites for scuba diving. Regarding biological preference, Table 5 demonstrated that the second model (diving experiences) could significantly predict divers' biological preferences after controlling the effects of sociodemographic variables. Diving experiences ($b = 0.11$, $p = 0.007$, $\beta = 0.15$) could explain an additional 2% of the variance of biological preferences, even after the effects of gender, age, education level, and salary were statistically controlled for ($\Delta R^2 = 0.02$, $F$ change (1, 392) = 7.34, $p = 0.007$). This indicates that diving experience can play a significant role in predicting divers' biological preferences when selecting sites for scuba diving.

**Table 5.** Coefficients of diving experience on divers' physical and biological preferences.

| Independent Variables | Physical Preferences | | | | | | Biological Preferences | | | | | |
|---|---|---|---|---|---|---|---|---|---|---|---|---|
| | *Model 1* | | | *Model 2* | | | *Model 1* | | | *Model 2* | | |
| | *b* | *SE(b)* | *β* | *b* | *SE(b)* | *β* | *b* | *SE(b)* | *β* | *b* | *SE(b)* | *β* |
| Gender | 0.11 | 0.07 | 0.08 | 0.11 | 0.07 | 0.08 | 0.15 | 0.09 | 0.09 | 0.15 | 0.08 | 0.09 |
| Age | 0.02 | 0.04 | 0.03 | −0.01 | 0.04 | −0.02 | 0.00 | 0.05 | 0.00 | −0.06 | 0.05 | −0.07 |
| Education | −0.04 | 0.06 | −0.04 | −0.06 | 0.06 | −0.05 | 0.02 | 0.07 | 0.02 | −0.01 | 0.06 | −0.01 |
| Salary | −0.04 * | 0.02 | −0.13 | −0.04 * | 0.02 | −0.13 | −0.05 | 0.02 | −0.13 | −0.05 ** | 0.02 | −0.14 |
| Diving experiences | | | | 0.06 | 0.04 | 0.10 | | | | 0.11 ** | 0.04 | 0.15 |
| R | | 0.16 | | | 0.18 | | | 0.16 | | | 0.21 | |
| R² | | 0.03 | | | 0.03 | | | 0.03 | | | 0.04 | |
| ΔR² | | 0.03 | | | 0.01 | | | 0.03 | | | 0.02 | |
| Std. Error | | 0.71 | | | 0.71 | | | 0.83 | | | 0.82 | |
| *df* | | 4 | | | 1 | | | 4 | | | 1 | |
| *N* | | 393 | | | 392 | | | 393 | | | 392 | |
| F-statistic | | 2.54 | | | 2.88 | | | 2.60 | | | 7.34 | |
| (F-statistic) | | 0.039 * | | | 0.091 | | | 0.036 * | | | 0.007 ** | |

Note: $p < 0.05$ *, $p < 0.01$ **.

## 5. Discussion

Several observations should be made in light of the study's results. First, this study confirmed that divers' sociodemographic characteristics, including age and education, had different levels of influence on the level of diving experience, which lends strong support to Hypothesis 1. These findings are consistent with previous research conducted in Malaysia and Italy that demonstrated that divers' age and education significantly impact their level of diving experience [31,45]. As divers become older, they may have more life experience and travel opportunities that allow them to engage in diving-related activities. Similarly, higher education may be associated with higher wages, which enable divers to experience less financial pressure when acquiring better diving equipment while receiving more professional training from diving courses to enhance their diving experiences.

In addition to diving experience, divers' physical and biological preferences were considered by analysing their sociodemographic characteristics. Results indicated that divers' monthly salary negatively predicted their physical and biological preferences. This confirms Hypothesis 2, indicating that divers with increased salaries perceived fewer physical and biological preferences in selecting diving sites in Hong Kong. The reason for explaining this phenomenon may be that scuba diving in Hong Kong is less promoted and publicized than in dive sites elsewhere. Divers with higher monthly wages may prefer to

dive abroad due to their subjective belief that good dive sites with excellent physical and biological environments generally exist overseas rather than in Hong Kong.

Finally, divers' physical and biological preferences were regressed with their level of diving experience, and corresponding separation results were derived from testing the third hypothesis. In terms of physical preferences, the diving experience of Chinese divers could not significantly predict physical preference in regard to site selection, which indicates that the level of divers' diving experience did not raise concerns about the physical conditions of diving sites. However, our findings challenge the results of Meisel-Lusby and Cottrell [13], who suggested that as the accumulation of diving experience increases, experienced divers perceive a clear preference for shipwreck enjoyment due to the company of their familiar friends. Moreover, Bentz, Lopes, Calado, and Dearden [53] found that experienced divers clearly preferred dive sites with uncontaminated, undamaged, and uncrowded underwater rock formations such as caves, arches, and seamounts in a study of divers' motivation in the Azores, Portugal. The discrepancy between this study and previous studies may be due to differences in the timing of data collection during the pandemic outbreak. The local authorities in Hong Kong employed various stringent measures to combat the pandemic, including social isolation, vaccination programmes, border closures, and quarantine, making domestic and international trips challenging during this period of travel restrictions. In contrast, previous studies of divers' preferences were not conducted during the pandemic outbreak, and the tourism industry was on the right track compared to the current recreational conditions in Hong Kong. For this reason, the criteria and expectations of divers with regard to their physical preference for dive destinations may not be as crucial as the thirst for recreation (despite the level of diving experience), especially given the less selective recreational opportunities caused by the pandemic.

With regard to biological preferences, our findings suggest that diving experience has a significant relationship with biological preference, meaning that more experienced Chinese divers prefer dive sites with rich biological conditions. This result is consistent with prior research conducted in Malaysia and Brazil that found that as divers gain more diving experience, they acquire a stronger preference for marine biodiversity [8,16]. However, these results contradict recent research conducted in South Africa by Lucrezi, Saayman, and van der Merwe [20], who stated that diving is about escaping, relaxing, and spending time with friends for enjoyment and suggested that diving preference is not influenced by diving experience. Equally, Lucrezi, Saayman, and van der Merwe [20] further indicated that scuba divers did not change their preferences for specific marine species and were equally excited about dolphins, sharks, turtles, and rays regardless of their diving experience. The discrepancy between the current and previous studies may be due to differences in the data analysis methods. In particular, the regression models were applied to our data analysis, while the previous research conducted by Lucrezi, Saayman, and van der Merwe [20] was performed using a structural equation model. As stated by Werner and Schermelleh-Engel [66], structural equation models can be integrated with different statistical methods, such as path analysis, factor analysis, and regression analysis, to address multiple research variables to finalize the research model. Conversely, multiple regression is based on the prior identification of the study variables from the relevant theoretical literature before allowing subsequent statistical testing. Therefore, the structural equation model yields more potential experimental results than multiple regression and is more resilient than regression in terms of the methodological analysis of the data [67]. For this reason, different results between the current and previous studies concerning the effect of diving experience on biological preferences may be presented due to possible differences in the data analysis methods. Therefore, a more comprehensive study of research variables and appropriate statistical methods should be considered for further research in understanding divers' diving preferences in Hong Kong.

## 6. Conclusions and Implications

This study first explored the relationship between divers' demographic characteristics concerning diving experience and diving preferences, followed by a synchronous analysis exploring diving preferences based on divers' level of diving experience. This study demonstrated that divers' sociodemographic characteristics were statistically related to their diving experience. Additionally, diving experience was found to be statistically associated with biological preference but not physical preference. These findings have not previously been systematically investigated in Hong Kong, especially among Chinese scuba divers. Under these circumstances, the results of our study established a theoretical framework for the association between divers' sociodemographic characteristics and dive experience on divers' particular dive preferences and confirm the significant relationship between dive experience and dive preferences in Hong Kong. These findings can provide a theoretical and practical contribution to developing sustainable tourism, especially nature-based tourism. Furthermore, these findings may help authorities develop appropriate strategies for visitor management practices in marine protected areas to enhance the sustainable development of scuba diving tourism while minimizing its negative environmental and ecological impacts in Hong Kong. Consequently, based on the findings discussed above, we offer some policy recommendations as references for relevant agencies and stakeholders in supporting visitor management for the continued success of scuba diving tourism.

First, the discussion noted that divers' degree of diving experience does not have a relationship with the physical preference of local dive sites. The reason for this phenomenon may be the lack of market share due to the lack of promotion of local diving markets combined with the pandemic's negative impact, resulting in fewer recreational opportunities, which lowered divers' physical preference for local scuba diving. Therefore, (1) we recommend that the Hong Kong government adopt effective marketing campaigns to promote scuba diving tourism by exploring dive sites other than the traditional ones already in the marine protected areas. This may ensure that the site environment does not homogenize, which could help support the development of scuba diving tourism in Hong Kong. In the meantime, the government should continue collaborating with relevant functional departments to monitor the pandemic by offering more diving opportunities for divers of all levels of diving experience, especially given the changes in the pattern and scale of the local tourism industry due to the current pandemic situation.

In addition to physical preferences, this study demonstrates that biological preference is the most influential component of divers' preferences, with coral communities being the most popular and influential feature to encourage divers' participation. Therefore, we urge the government of Hong Kong to employ a variety of environmental precautions to prevent the degradation of marine life, particularly coral ecosystems. For example, (2) coastal ecological management techniques should be established, including setback architecture restrictions, preservation of coastal vegetative cover, beach replenishment, and prohibitions on coastal infrastructure such as piers and seawalls [49]. Equally important, the relevant authorities should preclude divers from wearing gloves, engaging in illegal fishing, and delivering some environmental lectures before allowing them to travel to the marine protected areas [16,18]. Furthermore, the authorities should consider developing artificial reefs in appropriate marine protected areas to reduce the pressure on natural coral damage caused by increasing tourist visitation in Hong Kong. This refers to a previous study that demonstrated that artificial reefs serve as a very effective management measure to balance the pressure between recreational demands and the degradation of marine ecosystems [9,10,17,52]. Most importantly, implementing zoning and access restrictions by referring to the carrying capacity estimation may help in addressing marine conservation issues more appropriately [16,43,68].

Finally, the sociodemographic characteristics of divers were found to be related to diving experience and preferences to varying degrees. This suggests that understanding divers' preferences based on various demographic characteristics may assist tourism operators in developing diverse recreational opportunities to fulfil the specific recreational

market demands of divers while further enhancing the development of Hong Kong's tourism competitiveness. As a result, (3) we recommend that stakeholders should conduct more marketing research by offering a variety of diving and recreational programmes according to divers' sociodemographic characteristics and diving experience, such as performing market segmentation into commercial diving, technical diving, and cave and cavern diving since a positive relationship has been identified in the relationship between divers' sociodemographic characteristics and experience concerning their specific diving preferences [13,16,19,51]. Crucially, the success of these strategies depends on cooperation between stakeholders and authorities' engagement in formulating long-term sustainable management plans [5,40]. However, a lack of effective collaboration may result in difficulty in achieving and ensuring the sustainable development of the tourism industry and ecological protection [68].

## 7. Limitations and Future Research

Some research limitations have been identified and should be appropriately addressed to provide future studies with scientific suggestions to avoid research bias. On the one hand, the data collection method may have affected the overall response rate due to the small sample size, difficulties maintaining participants' confidentiality, and ethical issues under pandemic restrictions [62]. For this reason, we suggest that future studies integrate with other probability sampling methods to make the total sample size more representative and reliable [61]. On the other hand, as Ditton, Osburn, Baker, and Thailing [9] note, divers' preferences for particular social and psychological experiences are crucial to consider in addition to the local environment of dive sites. Thus, adding more appropriate variables may contribute to a deeper understanding of divers' specific diving preferences from a different perspective since the current variables may not be sufficient to understand diving preferences comprehensively. Notably, although previous studies have applied recreational specialization [53], motivation [19–21], and perception [16,17] to understand divers' diving preferences in different countries, these concepts have not been adequately applied in studying divers' diving preferences within the Chinese context in Hong Kong. Therefore, it is essential to ensure that research endeavours comprehensively examine scuba divers' participation preferences across all aspects of industrial development, focusing on marketing and tourist management development. Alternatively, further research findings should be shared with certifying agencies and key stakeholders to promote and maintain the significant growth and sustainability of diving-related activities in marine protected areas in Hong Kong.

**Author Contributions:** Conceptualization, L.T.O.C. and K.Z.; methodology, L.T.O.C. and K.Z.; validation, L.T.O.C., K.Z. and A.T.H.M.; formal analysis, L.T.O.C. and K.Z.; investigation, L.T.O.C., K.Z. and A.T.H.M.; writing—original draft preparation, L.T.O.C. and K.Z.; writing—review and editing, L.T.O.C., K.Z., T.W.L.L., W.F. and A.T.H.M.; supervision, L.T.O.C.; funding acquisition, L.T.O.C. All authors have read and agreed to the published version of the manuscript.

**Funding:** This research was funded by General Research Fund (GRF) of the Research Grants Council of Hong Kong: RGC Ref. No. 54218611919.

**Institutional Review Board Statement:** The study was conducted in accordance with the Declaration of Helsinki and approved by the Institutional Ethics Committee of The Education University of Hong Kong for studies involving humans.

**Data Availability Statement:** Available upon request.

**Acknowledgments:** The authors would like to extend their sincere gratitude for the funding support from the General Research Fund (GRF) of the Research Grants Council of Hong Kong (RGC ref. no. 18611919) and the student helpers from the Education University of Hong Kong for helping with the questionnaire surveys.

**Conflicts of Interest:** The authors declare no conflict of interest.

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
