# Peer review of "The Influence of Sociodemographic Characteristics and the Experience of Recreational Divers on the Preference for Diving Sites"

_sustainability, doi:10.3390/su15010447_

Round 1
Reviewer 1 Report
This is a good study. As I don't have any issues with the study design, my main concern will be its originality and the limited contribution to the body of knowledge. Various studies were done within the realm of scuba diving and Hong Kong. So what's new is missing.
Author Response
Please find attached the responses for reviewer's comments.
Thank you for the valuable comments

Reviewer 2 Report
Overall evaluation: This study explored the association between diving preferences and divers' social demographic characteristics as well as diving experiences through multiple regression analysis. The conclusions of this study are interesting and can provide references for tourism management. However, I believe some points need to be clarified and the writing needs to be improved further. Here are some of my concerns.
1. Keywords can be reduced and only the most important ones are mentioned, for example, the MPAs can be deleted.
2. The article format needs to be modified according to the journal's requirements, and authors can download the template from the journal's website.
3. Introduction: In order to expand the universal value of the article, author should first explain the need for the study of the topic chosen in the general context, and then introduce the situation about Hong Kong.
4. Literature review: Lines 64~67 mentioned the research objectives, however, this part did not do a good job of developing an analysis based on the objectives of the study. I believe there is a need to add “the influence of socioeconomic status on diving experiences” in literature review.
5. Policy implications need to be articulated. For example, the manuscript mentioned that “we recommend that the Hong Kong government adopt effective marketing campaigns to relaunch tourism development……”, so how to expand marketing from the specific behavioral level. Author needs to explain clearly in the manuscript.
Author Response
Please find attached the responses to reviewer's comments. Thank you for the valuable comments

Reviewer 3 Report
This manuscript deals with an interesting topic and addressed to aims and scope of this journal.
The manuscript is well written and well structured.
The approach is in line with the research aim and the methodology is appropriate. The results are clear and meet the proposed study aims.
Finally, the conclusions are consistent with the evidence and arguments presented.
Nevertheless, the authors should improve the link with sustainability which is the focus of this journal. Also, not all the references are accordingly to the format request. The authors must check all the references carefully and correct them providing complete bibliographic details of some references.
Author Response
Please find attached the responses for the reviewer's comments. Thank you for the valuable comments

Round 2
Reviewer 1 Report
Thank you for the revision.